# Nutritional and Therapeutic Properties of Fermented Camel Milk Fortified with Red *Chenopodium quinoa* Flour on Hypercholesterolemia Rats

**DOI:** 10.3390/molecules27227695

**Published:** 2022-11-09

**Authors:** Mohamed Saleh Al-Anazi, Khaled Meghawry El-Zahar, Nourhan Abdel-Hamid Rabie

**Affiliations:** 1Department of Food Science and Human Nutrition, College of Agriculture and Veterinary Medicine, Qassim University, Buraydah 51452, Saudi Arabia; 2Food Science Department, Faculty of Agriculture, Zagazig University, Zagazig 44511, Egypt

**Keywords:** camel milk, red quinoa, oxidative stress, lipid profile, liver and kidney functions, atherogenic index, obesity, adipocyte size

## Abstract

Quinoa is a nutrient-dense food that lowers chronic disease risk. This study evaluated the physicochemical and sensory qualities of fermented camel milk with 1, 2, 3, and 4% quinoa. The results showed that improvement in camel’s milk increased the total solids, protein, ash, fiber, phenolic content, and antioxidant activity more effectively. Fermented camel milk with 3% of quinoa flour exhibited the highest sensory characteristics compared to other treatments. Fermented camel milk enriched with 3% red quinoa flour was studied in obese rats. Forty male Wistar rats were separated into five groups: the first group served as a normal control, while groups 2–4 were fed a high-fat, high-cholesterol (HF)-diet and given 2 mL/day of fermented milk and quinoa aqueous extract. Blood glucose, malondialdehyde (MDA), low-density lipoprotein (LDL), cholesterol, triglyceride, aspartate transaminase (AST), alanine transaminase (ALT), alkaline phosphatase (ALP), creatinine, and urea levels decreased dramatically in comparison to the positive control group, while high-density lipoprotein (HDL), albumin, and total protein concentrations increased significantly. Fortified fermented camel milk decreased the number of giant adipocytes while increasing the number of tiny adipocytes in the body. The results showed that the liver and renal functions of hypercholesterolemic rats were enhanced by consuming fermented milk and quinoa. These results demonstrated the ability of quinoa and camel milk to protect rats from oxidative stress and hyperlipidemia. Further studies are needed to clarify the mechanisms behind the metabolic effects of fermented camel milk and quinoa.

## 1. Introduction

Mammalian milk is nutritious for new-borns. Camel (*Camelus dromedarius* L.) milk is especially popular in the Arabian Gulf. Camel milk differs significantly from cow’s milk in chemical components and coagulability. Beneficial uses of camel milk in the metabolic syndrome (which is the combination of abdominal obesity, dyslipoproteinemia, and hypertension with peripheral insulin resistance) have frequently been postulated [1]. Short-and long-term frequent ingestion of camel milk benefits hypertension and diabetes in adults, due to the inhibition of oxidative and inflammatory stressors [2].

South American quinoa (*Chenopodium quinoa* Willd.) is a nutrient-rich grain eaten worldwide. It is one of the most flexible grain foods, with flour, flakes, and cereals as well as commercial or packaged products. Quinoa has a nutritional benefit that is comparable to that of meat and greater than that of other cereals. Quinoa is not extensively consumed, notwithstanding it has great nutritional values, for a variety of reasons, including high import prices as well as a lack of market knowledge of its advantages [3]. Quinoa’s dietary fiber lowers blood sugar and cholesterol levels after a diet even while controlling cholesterol synthesis in the liver [4]. Quinoa seeds provide more essential amino acids, protein, carbohydrates, and fiber than other grains. Those who chew a high-fat, high-cholesterol diet may benefit from increasing protein consumption to prevent hyperlipidemia [5]. Quinoa seeds, especially red seeds, have a high proportion of phenolic and flavonoid antioxidants [6]. Furthermore, because the glycosaminoglycan portion of this dietary fiber contains emulsifying and stabilizing qualities, it has great potential for future use in the food industry [7]. Quinoa seeds can be used as an alternate source of nutrients, especially protein, while enhancing polyphenol, antioxidant, and microbial activity [8].

However, information in the literature on camel milk structure clearly varies, depending on variables, such as the geographical origin of the milk studied, race, temporary or physiological variants, keeping (especially feeding) conditions, animal health status or biological stage, and analytical weaknesses [9]. Camel milk contains all the essential nutrients on which lactic acid bacteria can readily act and produce many biofunctional components that give health benefits to the consumer. Additionally, an increase in the concentration of amino acids, fatty acids, and organic acids was observed after the fermentation of camel milk [10]. Hailu et al. [11] provided some information on camel milk proteins, focusing on major differences in protein composition and molecular characteristics.

The literature describes the biofunctionalities of camel milk and its protein hydrolysates, such as antioxidants, antidiabetic, anticancer, angiotensin-converting enzyme inhibitory, antimicrobial, anti-inflammatory, hepatoprotective, antiradical, antiallergic, and anti-autistic effects. Some micronutrients, antioxidant enzymes, and protective proteins are biologically significant [12]. Potential health benefits include inhibition of hypertension-conversing enzymes, antimicrobial and antioxidant characteristics, and antidiabetic activities. Hyperlipidemia and hypercholesterolemia may be responsible for modifying and increasing LDL, free radicals, and lipid peroxidation components, which are the main risk factors for heart disease [13]. The bioactive peptides, such as ACE-inhibitory peptides and antioxidative peptides, were also derived from fermented camel milk, and the comparative study also showed the maximum ACE-inhibitory in fermented camel milk rather than bovine milk [14,15]. Moslehishad et al. [14] demonstrated that fermentation of camel milk with single strains of lactic acid bacteria (LAB) generated antioxidant activity in the whey fractions. Lactic acid bacteria produce peptides from milk proteins during fermentation and provide several health benefits to consumers. 

Radical scavenging is the main mechanism by which antioxidants act in foods. Due to the adverse effects of current cardiovascular disease treatments, such as gastrointestinal difficulties, hypertension, excitability of liver enzymes, and liver failure, as well as their exorbitant price, they are not a simple response [16]. Lactic acid bacteria have been shown in experimental animals and humans to reduce blood cholesterol levels. Its ability to decrease cholesterol is probably due to its lack of binding to bile salts by certain bacteria strains producing the enzyme bile salt hydrolase [17,18]. The bacteria could secrete bile salt hydrolase, which could successfully reduce cholesterol levels by increasing the release of bile salts and then reducing absorption in the intestine [19,20]. The purpose of this study is to determine the nutritional and therapeutic effects of fermented camel milk supplemented with red quinoa flour on sensory properties, lipid profile, liver and kidney functions, atherogenic index, and histopathological characteristics in hypercholesterolemic rats.

## 2. Results and Discussion 

### 2.1. Chemical Composition and Phytochemical Properties of Red Quinoa Seed

Chemical compositions and phytochemical capabilities of quinoa seeds are presented in Table 1a–c. Red quinoa moisture, crude protein, dietary fiber, ash, crude fat, and carbohydrate content were 10.22%, 15.03%, 8.8%, 3.22%, 5.2%, and 57.6%, respectively (Table 1a). These findings are consistent with the data obtained in previous studies by Navruz-Varli et al. and Diaz-Valenica et al. [21,22], who detected that the fat, protein, ash, and dietary fiber contents of red quinoa were 7.33, 14.0, 2.51, and 14.30 g/100 g, respectively. Ca, P, Mg, Na, and K were the most abundant minerals found in red quinoa flour, with concentrations of (148.7, 383.9, 246.9, 12.2, and 926.7 mg/100 g^−1^), respectively (Table 1b). In addition, the concentrations of Fe, Mn, Cu, and Zn were found to be 13.2, 10, 5.1, and 4.4 mg/100 g^−1^, respectively, which is comparable with the findings of Abugoch-James et al. [23]. These findings are in accordance with those obtained by Diaz-Valencia et al. [22], who discovered that red quinoa contains 699, 578, 73, 17.5, 243, 5.63, 3.70, and 0.47 mg/100 g^−1^, of K, P, Ca, Na, Mg, Fe, Zn, and Cu, respectively. 

Total phenolic compounds, total flavonoids, and vitamin C were all found in higher concentrations in the red quinoa seeds (488.7, 367.1, and 16.4 mg/100g DW) respectively. Red quinoa seeds showed higher levels of total phenolic compounds, total flavonoids, and vitamin C (488.7, 367.1, and 16.4 mg/100g DW), respectively (Table 1c). Furthermore, quinoa’s antioxidant activity and radical scavenging activity are strongly linked to the presence of phenolic components [24,25]. Sampaio et al. [26] found that quinoa seeds have stronger antioxidant potential than entire grains (wheat, barley, millet, rice, and buckwheat), indicating that quinoa contains potent free radical scavenging components. Therefore, quinoa appears to be a promising candidate for the development of new treatments and therapeutic options for disorders related to oxidative stress. 

### 2.2. Gross Chemical Composition and Sensory Evaluation of Different Fermented Milks 

The pH and titratable acidity values in camel milk were found in the range of 6.22 to 6.56 and 0.13 to 0.25% *w*/*v*, respectively. The percentages of fat, protein, ash, lactose, dry matter content, and the density of the camel’s milk samples were as follows: fat 3.28, protein 3.10, ash 0.83, lactose 5.23, and total solids 11.83, respectively (data not shown). 

The results showed that the fat content in fermented milk increased slightly in comparison with fresh camel milk. The amount of lactic acid produced increased with a concomitant drop in pH with an increase in fermentation time. However, the combination of *L. bulgaricus* and *S. thermophilus* (1:1) resulted in a lower pH and higher acidity. The total protein decrease in fermented milk samples occurs as a result of protein hydrolysis by the starter culture. Protein degradation in all treatments increased due to the limited hydrolysis of milk proteins by lactic acid bacteria. Fermented milk was produced by using a traditional starter culture and then evaluated for organoleptic quality and consumer acceptance. The ideal fortifying ratio for camel milk with red quinoa was (3% *v/w*), which was used in the biological study to estimate the impact of red quinoa, as well as camel milk, on the blood lipid profile in hyperlipidemic rats. The amount of lactic acid produced increased with a concomitant drop in pH with an increase in fermentation time. The pH of fermented camel milk containing red quinoa seed flour (Cam_Q_4_, Cam_Q_3_, Cam_Q_2_ and Cam_Q_1_, respectively) was higher than that of camel milk (Table 2a). Furthermore, fermented camel milk fortified with quinoa flour exhibited a higher acidity than fermented camel milk. These findings are consistent with the results acquired by Al-Harbi et al. and El-Zahar et al. [27,28], which indicated a rise in acidity when stored at the refrigerator temperature. The protein, fat, fiber, and ash content of the fermented milk rapidly increased when the replacement ratio was increased as a result of fortifying camel milk with quinoa seed flour by approximately 8.8%, 10%, 46%, and 18.7%, respectively. This could be due to the higher protein and carbohydrate content of quinoa compared to camel milk [22]. Protein hydrolysis by starting culture resulted in a decrease in total protein in fermented milk samples. These findings are consistent with those reported by Omar et al. [29], who discovered that partially replacing camel milk with preserved skim milk increased protein, ash, and carbohydrate content in the bio-yogurt produced. It was also found that adding barley bran to yogurt increased the protein, ash, and carbohydrate content [30]. Adding more quinoa flour to fermented camel milk increases the amount of total phenolic components compared to the control. The total phenolic content and radical scavenging activity of fermented camel milk were increased in camel milk fortified with varying percentages of oat, kiwi, and avocado [24,31]. 

The mean values of sensory evaluation scores are shown in Table 2b. Fermented camel milk had lower sensory scores for body and texture than fermented camel milk enriched with quinoa (Cam_Q_3_, Cam_Q_2_, Cam_Q_4_ and Cam_Q_1_, respectively). The overall acceptance scores of the sensory attributes revealed that the fermented milk with 3% of red quinoa was acceptable, followed by the fermented milk fortified with 2%, while the fermented milk fortified with 4% was the lowest ranked. The data in Table 2 shows that the fat and protein content, as well as the overall sensory quality of the camel’s milk fermented with 3% of quinoa flour (Cam_Q_3_), were significantly higher (*p* ≤ 0.05) than that of the fermented camel milk (Cam_T). High-saturated fatty acid content, salts, weak lipolysis, and poor protein structure can all contribute to a low sensory rating of fermented camel milk [32]. Improvements in the sensory quality of fermented milk were mostly due to the development of starter cultures throughout the fermentation or storage stage. The findings of this investigation were in accordance with those of [28,29,31]. Curti et al. [33], found that the addition of higher concentrations of quinoa flour into yogurts formulations resulted in reduction in acceptability of aroma and flavor. Although the colour of the final product was darkness and thus not so appealing but, sensory colour values of Kishk-soup samples containing quinoa seeds were not significantly different and most of the soups samples were accepted in position of sensory colour values and have the potential to be well received by consumers. 

### 2.3. Effect of Fermented Milk on the Body Weight Gain in Rats

The effect of oral administration of different fermented milks (2 mL/day) on rat weight given a HF-diet was demonstrated in Figure 1. It was shown that fermented milk contributed to a drop in blood fat levels and relative weight in rats. The initial body weights of all rat groups were significantly higher when compared to the negative group. Whereas the final body weights of all rat groups were not significantly different, with the exception of the positive control group, which was significantly higher (*p* ≤ 0.05). Furthermore, these findings are in accordance with Halaby et al. [34], who discovered that a diet fortified with 30% and 40% quinoa seed powder can improve body weight gain, feed consumption, and feed efficiency ratio. Our findings are corroborating those previously released reports by [28,29]. It may be concluded that the addition of fermented camel milk fortified with red quinoa seed flour in the rats’ diet resulted in a significant increase (*p* ≤ 0.05) in the final weight and body weight compared with the other groups. 

Data are presented as the mean ± SD (*n* = 3/group). NC) nontreated nonhypercholesterolemic rats (negative control), PC) hypercholesterolemic rats (positive control), Cam_T) hypercholesterolemic rats treated with fermented camel milk, Cam_Q_3_) hypercholesterolemic rats treated with fermented camel milk fortified with 3% red quinoa seed flour and EX_Q_3_) aqueous solution of red quinoa seeds flour.

### 2.4. Effect of Fermented Camel Milk on Fasting Serum Insulin and Glucose Levels in Obese Rats

Untreated rats feeding on HF-diet had a significant increase (*p* < 0.05) in fasting serum insulin and glucose levels by 13.3 mlU/L and 149.2 mg/dL, respectively, when compared to the normal group rats (8.4 mlU/L and 66.9 mg/dL). Animal protection with fermented camel milk and quinoa seed significantly lowered serum insulin and glucose levels when compared to the positive control group (Figure 2). Additionally, blood insulin and glucose levels were normalized in rats fed fermented camel milk. An increase in muscle triglyceride content and visceral fat deposition contribute to hyperlipidemia and hyperinsulinemia, and the HF-diet is reported to promote insulin resistance [28]. The HF-diet increased body weight and caused hyperinsulinemia, which was linked to hyperglycemia and hyperlipidemia, according to our study results. This could be because quinoa seeds contain more phenolic compounds and fibers, which stimulate pancreatic cells to release insulin [34].

Data are presented as the mean ± SD (*n* = 3/group). NC) nontreated nonhypercholesterolemic rats (negative control), PC) hypercholesterolemic rats (positive control), Cam_T) hypercholesterolemic rats treated with fermented camel milk, Cam_Q_3_) hypercholesterolemic rats treated with fermented camel milk fortified with 3% red quinoa seed flour and EX_Q_3_) aqueous solution of red quinoa seeds flour.

### 2.5. Effect of Fermented Camel Milk on Fasting Serum Insulin and Glucose Levels in Obese Rats

Serum concentrations of triglyceride (TG), total cholesterol (TC), LDL and very low-density lipoprotein (VLDL) were all significantly reduced, while HDL was significantly increased, when red quinoa seed was supplemented with fermented camel milk and given to rats that had been subjected to oxidative stress. Table 3 reveals that TC values in the positive control group (154 mg/dL) rose significantly compared with the negative control group (64 mg/dL), whereas TC levels were 105, 96, and 99 mg/dL, respectively, in rat doses with Cam_T, Cam_Q_3_, and EX_Q_3_. A rise in HDL levels is a useful indicator for anti-hypercholesterolemia drugs. These grains’ dietary fiber can inhibit cholesterol absorption and bind to bile acid, enhancing cholesterol catabolism in the colon and creating short-chain fatty acids, reducing liver cholesterol synthesis [4,35]. Foucault et al. [36], possessed the capability of quinoa to reduce serum glucose, TG, and LDL levels in rats fed a fructose-enriched diet, consequently preventing the deleterious effects of fructose on HDL. Quinoa may guard against peroxidation by raising antioxidant capacity and reducing oxidative damage in rat plasma and tissues. Generally, feeding with fermented camels fortified with red quinoa significantly reduced LDL levels (41.2 mg/dL) compared to the positive control group (98.5 mg/dL). The LDL value was reduced from (53, 41, and 49.2 mg/dL) in the Cam_T, Cam_Q_3_, and EX_Q_3_ groups, respectively. This investigation corroborates the prior findings [27,28]. Fermented camel may decrease cholesterol and triglyceride levels by many proposed mechanisms: (1) inhibition of liver cholesterol synthesis and distribution of cholesterol from the blood circulation to the liver, (2) intestinal bacteria may inhibit the absorption of cholesterol, and (3) milk fermented by lactic acid strains may inhibit cholesterol synthesis enzymes and decrease cholesterol levels [37,38]. Moreover, bioactive peptides derived from camel milk proteins may reduce cholesterol through the interaction between bioactive peptides and cholesterol, and the presence of orotic-acid in camel milk, which is thought to be responsible for lowering cholesterol levels in rats [28]. 

### 2.6. Effect of Fermented Camel Milk Fortified with Red Quinoa on Serum AST, ALT, ALP, Liver, and Kidney Functions in Obese Rats

Hypercholesterolemia has been known to disturb the oxidant-pro-oxidant balance and reduce the efficacy of the antioxidant protection system, leading to tissue damage and frequently correlated with the evolution and development of atherogenesis [39]. Lipid peroxidation is an oxidative modification of polyunsaturated fatty acids in the cell layers that produces several degeneration products. Certain products cause great disturbance to cell elements and may prevent cell replication and cell endurance. The activities of AST, ALP, and ALT were evaluated in blood serum as liver function indicators. As shown in Table 4, the impact of the treatments is described above on the levels of AST, ALT, and ALP. Generally, the negative control group showed significantly lower serum AST, ALT, and ALP levels (25, 105, and 52 U/L) in contrast to the hypercholesterolemia rat group (59, 180, and 115 U/L). Overall, given hypercholesterolemic rats, fermented milk fortified with red quinoa reduced the AST, ALP, and ALT levels compared to the positive control group, where AST levels decreased from (80, 73, and 84.3 mg/dL) in the Cam_T, Cam_Q_3_, and EX_Q_3_ groups, respectively. The presence of ascorbic acid and phenolic components, as well as flavonoids known as hepatoprotective factors, may explain the liver’s activity [35]. The high content of camel milk antioxidants, which preserve the plasma membrane of hepatocytes and protect them from rupture and exit of the cytosol loaded with these enzymes, can be attributed to a decrease in transaminase enzymes and the recovery of hepatocytes for some of their vital functions [28]. Rats that fed on fermented camel milk containing 3% red quinoa flour improved, which may be due to the chemical structure of quinoa seed, which has antihepatotoxic and antioxidant activities [5,6]. Table 4, shows that there are significant changes in either albumin or total protein due to feeding on various fermented milks, with albumin values ranging from 3.7 to 4.15 g/dL, and total protein from 7.3 to 7.9 g/dL, respectively. Albumin values increased by 3.8, 4.15, and 3.7 mg/dL in treated rats with different treatments (Cam_T, Cam_Q_3_, and EX_Q_3_), respectively. Lactic acid’s ability to control blood lipid alterations could explain the rise in total protein and albumin levels. This reduces liver tissue damage caused by high blood lipids and cholesterol. The results indicated that fermented camel milk containing probiotic bacteria increased protein and total albumin, and that the effect was related to the starting culture [27]. Urea and creatinine values have risen significantly (*p* ≤ 0.05) in rats feed with HF-diet (40.9 and 2.4 mg/dL) compared to the negative control group (17.5 and 0.89 mg/dL) respectively. Generally, ingesting fermented milk containing both probiotics and traditional fermented milk was effective at lowering creatinine and urea levels [26]. Martinez et al. [40], found that a large increase in creatinine concentration in blood serum in rats fed an HF diet raises the risk of renal injury. 

The improvement in liver function may result from feeding groups of rats, either conventional fermented milk or probiotics, and this was confirmed by the histological examination of the liver (data not shown). At the end of the experiment, hepatic tissue appeared almost natural without congestion or cellular infiltration. In the kidney, creatinine is distilled by the glomerulus and excreted by the tubules, and only free creatinine appears in the blood serum. The reason for enhanced serum enzymes and renal functions in HF-diet-induced liver and renal injury by camel milk may be due to the prevention of lactic acid and antioxidant compound abilities.

### 2.7. Effect of Fermented Camel Milk Fortified with Red Quinoa Seed Flour on the Activity of Serum MDA, GSH, SOD and TAC Level Enzymes in Hypercholesterolemic Rats

The data illustrated in Table 5, show that the obese rats had significantly lower serum glutathione peroxidase (GSH-px) and superoxide dismutase (SOD) levels as well as reduced MDA value compared to the control rat group. Hypercholesterolemic rats that were fed on fermented camel milk, fermented camel milk fortified with red quinoa, and aqueous extract of red quinoa seeds had significantly higher serum GSH-px and SOD levels and lower serum MDA levels compared with the positive control group. These advancements demonstrated that the fermented camel milk had antioxidative and health benefits for the liver recovering from HF-diet injury. These enzymes rose considerably, suggesting camel milk and red quinoa have anti-hepatotoxic and antioxidant effects [41]. Ascorbic acid and the phenolic components present in camel milk and quinoa may be explained by their ability to scavenge free radicals and active oxygen species [42]. Quinoa supplementation lowered plasma malondialdehyde levels and improved antioxidant enzyme activity in oxidative stress-induced rats [43]. 

Polyphenols and flavonoids contained in red quinoa seeds may be helpful in decreasing oxidative stress and liver inflammation as bioactive substances. Li et al. [43], explored the potential of quinoa to decrease peroxidation and maintain antioxidant enzyme activity, as well as its capacity to control lipid oxidation hydroxyl radicals. Another suggested mechanism is quinoa seed’s potential to raise GSH-px and thus rectify cardiac cells’ inadequate thiol state. 

### 2.8. Effect of Fermented Camel Milk Fortified with Red Quinoa on Adipose Tissue Weight and Adipocyte Volume in Hypercholesterolemic Rats

Liver and white adipose tissue weight, and adipocyte cell size are presented in Table 6 and Figure 3a–d. The results showed a significant difference between the NC and PC groups in the weight of adipose tissue and adipocyte size. There were no significant differences between the NC (Figure 3), Cam_T (Figure 3c), Cam_Q_3_ (Figure 3b), or EX_Q_3_ (Table 6) groups in either adipose tissue weight or adipocyte cell size, even though rats in these groups were fed on a HF-diet (Figure 3b). Thus, all of these treatments (Cam_T, Cam_Q_3_, and EX_Q_3_) had a significant effect in decreasing adipose tissue weight and adipocyte size (*p* ≤ 0.05). The combined effect of camel milk and quinoa seed flour on obesity in a high-fat-fed rat model was examined for the first time in the present study. These beneficial effects were associated with a significant reduction in the body weight gain of the rats (Figure 2). In contrast to the rats’ feed with HF-diet, the fermented camel milk and fermented camel milk fortified with quinoa seed flour significantly reduced the adipocyte size in rats, as shown in Table 6. The reduction in adipocyte size was a result of the significant reduction in the adipose tissue mass of the experimental groups compared to the positive control group (Table 6). These results are in line with other studies that have reported a reduction in adipocyte size [44], and adipose tissue mass [45]. 

### 2.9. Histological Changes in Experimental Rats Fed on Fermented Camel Milk Fortified with Red Quinoa 

The normal histological structure of the central vein and surrounding hepatocytes in the parenchyma was observed in histological analyses of liver sections from rats fed on a standard diet (Figure 4). Arterial degradation was detected in the cytoplasm of hepatocytes in the hypercholesteremia rat group, as well as pyknosis in the nucleus of several other hepatic cells and inflammatory cell infiltration in the portal area. Furthermore, the portal area showed hyperplasia, newly developed bile ducts, and portal vein dilatation (Figure 4b). The current study findings are consistent with those of Saleem et al. [46], who found that members of the Chenopodiaceae family, have significant hepatoprotective and antioxidant activity due to high concentrations of phytochemical compounds such as flavonoids and phenolic acids (Figure 4c,d). Saleem et al. [46], discovered that Chenopodiaceae family members exhibit strong cardio protective and antioxidant effects due to high quantities of phyto-constituent’s components, such as flavonoids and phenolic acids. These findings support previous findings [2,31,41] indicating quinoa seeds can improve liver function in rats on a high-cholesterol diet due to their high bioactive component content. The nephrons and tubules in the cortex of rats given a standard diet (negative group) revealed no histopathological alterations, and their histological structure was normal (Figure 4a). The animals fed an HF-diet, exhibited deterioration in the tubular lining epithelium as well as necrobiosis alteration in a distributed manner. The histopathological diagnosis revealed severe hepatotoxicity in HF-diet-fed rats. The present results agree with a previous report [47]. The biomarkers in the serum further confirmed the histopathological evaluation (Figure 4a–d).

The glomerular tufts demonstrated atrophy, edema, and degeneration in the corticomedullary lining tubular epithelium as well as congestion in the cortical blood vessels (Figure 5b). Furthermore, no histopathological alterations were observed in the kidneys of rats exposed to fermented camel milk and red quinoa seed flour, as shown in Figure 5c,d. Gentamicin GM-induced renal histological anomalies, such as degeneration of glomeruli and tubules, were suppressed by camel milk and showed better progress. The present study confirms that pre-treatment with camel milk attenuates GM unwanted, induced renal dysfunction, and cellular damage [48]. Certain quinoa vitamins’ bioactive constituents are important because they act as antioxidants in kidney cell membranes, such as selenium, magnesium, phyto-sterols, folic acid, and tocopherols, which are hypothesized to have antioxidant anti-carcinogenic qualities or operate as anti-inflammatory agents.

## 3. Material and Methods

### 3.1. Sample Collection

Fresh camel milk was obtained from the frame of College of Agriculture and Veterinary Medicine, Qassim University, KSA. Dried red quinoa seeds were obtained from a local market in Buraidah city, KSA, and subjected to grinding to obtain a fine powder. 

### 3.2. Chemicals 

Gallic acid, 1,1-diphenyl-2-picrylhydrazyl (DPPH), other chemicals, and reagents were purchased from Sigma-Aldrich (St. Louis, MO, USA). Kits for determination of serum albumin, total protein, TG, TC, HDL, ALT, AST, GSH-px, GSH-Rd, SOD, urea and creatinine were purchased from Human, Gesellschaft für Biochemical and Diagnostica mbH, Wiesbaden, Germany. 

### 3.3. Experimental Animals

Forty male Wistar rats (160 ± 10 g) were purchased from the College of Pharmacy, Qassim University, and divided randomly into five groups (8 rats/group). Under a 12-h light/dark cycle, the rats were housed in a specific pathogen-free environment at 21 ± 1 °C and 40–60% humidity. Rats were fed on a basal diet during the experimental period according to AIN-93 guidelines [49].

### 3.4. High Fat, High Cholesterol Diet (HF-Diet)

High-fat and high-cholesterol diet was prepared, containing (67 g standard diet, 31.70 g animal fat, 1% pure cholesterol, and 0.30% bile’s acid) [50].

### 3.5. Red Quinoa Flour Preparation and Aquoes Extract

Quinoa flour was prepared according to [23] with some modifications to remove saponins. Whole seeds were washed twice with cold water then seeds were soaked in the alkaline solution for 20 min, and then rinsed with 1% citric acid solution for 10 min. The cleaned seeds were washed with water until there was no foam as an indication of saponins removal from the seeds hull. Later saponins-free seeds were overnight oven-dried at 45 ± 1 °C. During drying treatment, the seeds were spread in a thin layer to avoid germination process and any further contamination. Finally, treated whole seeds were ground into flour using blender with miller (Hanabishi-HHANDBL10-3in1) and kept at 5°C for further analyses. The aqueous extract was prepared according to [51]. 50 g of red quinoa flour was mixed with 1000 mL distillate water (1:20 *w*/*v*) in 2L-glass beaker, and stirred for 60 min at room temperature. The extract was filtered through filter paper (Wattman No.1). The aqueous extract was stored at (5 ± 1 °C) for further analyses.

### 3.6. Starter Cultures and Fermented Milk Manufacture

The lyophilized Yo-flex culture (YC-x11) was obtained from Christian Hansen (Copenhagen, Denmark), contains a mixed strain of *S. thermophilus* and *L. bulgaricus* at the ratio (1:1). Different fermented milk was prepared according to the methods described by Tamime and Robinson [52]. Briefly, raw camel milk was heated at 90 °C/5 min then cooled to 42 °C. Quinoa seed powder was added at the rate of 1, 2, 3, and 4% immediately after incubation with 2% activated starter culture to avoid the quick reduction of pH and transferred into 100 mL plastic containers, lightly sealed, and incubated at 42 °C until the complete curd formation, then stored at refrigerator (5 ± 2 °C) for 24 h. The fermentation of yogurt was stopped when the pH value reached 4.6 as shown in Figure 6. 

### 3.7. Organoleptic Properties

Sensory studies were conducted with 20 students and lecturers of the Food Science and Human Nutrition department, College of Agriculture and Veterinary Medicine, Qassim University. Panellists were requested to evaluate the color (10 points), flavor (30 points), acidity (10 points), body and texture (40 points), and overall acceptability (10 points) according the scoring described by Tamime and Robinson [52].

### 3.8. Chemical Analysis

The chemical analysis was conducted on different fermented milk and red quinoa flour samples to determine the pH, titratable acidity, ash, protein, fiber, moisture, fat, vitamin C, and mineral content according to AOAC, [53]. The total polyphenolic content of quinoa seeds was measured by Folin-Ciocalteu assay using spectrophotometer (Secomam, France) based on the Singleton et al. [54] assay. The total flavonoid content of quinoa seeds was determined by the aluminium chloride colorimetric method as described by Seasotiya et al. [55]. The antioxidant activity was evaluated by the DPPH assay [56]. The scavenging antioxidant activity percentage was determined as follows: AOA(%) = 1 − Abs_sample_ − Abs_blank_/Abs_control_ × 100(1)

Total cholesterol, HDL, and triglyceride levels in serum were estimated by the method of [56]. Liver enzymes (ALP, ALT, and AST), serum albumin, and total protein were estimated as described by [57]. Urea and creatinine were determined as indicators of kidney function [58]. Insulin and blood glucose were determined in human blood samples according to the method of Thomas et al. [59]. Lipid peroxides were determined in serum as malondialdehyde MDA according to Namıduru et al. [60]. 

### 3.9. Experimental Design 

This study was conducted with the approval of the National Committee of Bioethics NCBE at King Abdul-Aziz City for Science and Technology, KACST, KSA, Review Board Number: 10024640, (Expiry date: 8 August 2024). Rats were randomly divided into five groups (*n* = 8). The first group served as normal control (NC) without further treatment. The other rats were switched to a HF-diet for 6 weeks and were subdivided into subgroups as follows: group (2) continued on the HF-diet without treatment and was labelled as a positive control (PC). Groups 3 to 5 were treated orally by intestinal tube with 2 mL/day of either fermented camel milk with traditional culture (Came_T), fermented camel milk fortified with 3% of red quinoa flour (Came_Q_3_), or red quinoa aqueous extract (Ex_Q_3_). Serum samples were obtained after the slaughter of rats at the end of the experiment by centrifuging the collected blood at 3000 rpm/10 min and stored at −20 °C until analysis.

### 3.10. Measurement of Adipocyte Size

The volumes of adipocytes are measured according to the method of Althwab et al. [45], in visceral fat surrounding the kidney, such as mesenteric, retroperitoneal, and epididymal white adipose tissue. The adipose tissues were rinsed with saline, and fixed in neutral 10% formalin solution, then embedded in paraffin, cut into 10 mm sections and stained with hematoxylin. Then the cell volumes were measured using the US National Institutes of Health image software (100 cells/mouse). 

### 3.11. Histopathological Examination

Autopsy samples were taken from the liver and kidney of the sacrificed rats and fixed in 10% formalin saline solution for ten hours at least, then washed in tap water for 12 h. Tissue specimens were cleared in xylene and embedded in paraffin. The obtained tissue sections were collected on the glass slides and dyed with hematoxylin and eosin stain for histopathological examination by the light microscope [61].

### 3.12. Statistical Analysis

All experiments as well as related analysis results were performed in triplicate and are presented as mean ± standard deviation. A variance analysis (two-way ANOVA) with a significance threshold of *p* ≤ 0.05 was used to define the differences between groups. Statistica 12.5 software was used to conduct the analysis (Stat Soft Inc., Tulsa, OK, USA).

## 4. Conclusions

In conclusion, red quinoa seed and camel milk can be used as effective food supplements for preventing overweight and obesity by lowering weight gain, adipocyte size, and serum levels of glucose and lipids. The fortification of camel milk with red quinoa seed powder improved the chemical, antioxidant, rheological, and sensory properties of fermented camel milk, and these improvements were proportional to the fortification up to 3%, which added nutritive and healthy benefits to the resultant fermented camel milk. Consumption of fermented camel milk containing 3% red quinoa seed flour in the hypercholesterolemic rat group caused a significant decrease in the levels of blood glucose, MDA, LDL, TC, TG, AST, ALT, ALP, creatinine, and urea and increased HDL, total protein, and albumin in comparison to hypercholesterolemic rats. Consequently, quinoa can be addressed as a substitute source of nutrients and especially protein, for people with dietary protein deficiency.

## Figures and Tables

**Figure 1 molecules-27-07695-f001:**
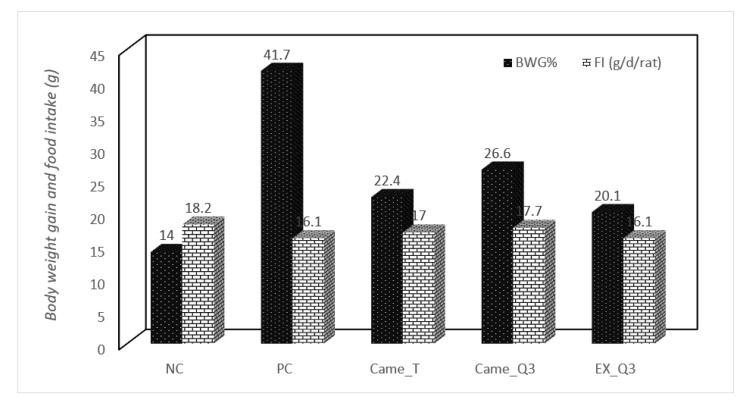
Effect of different fermented camel milks fortified with red quinoa seed powder on body weight gain and food intake in hypercholesterolemic rats.

**Figure 2 molecules-27-07695-f002:**
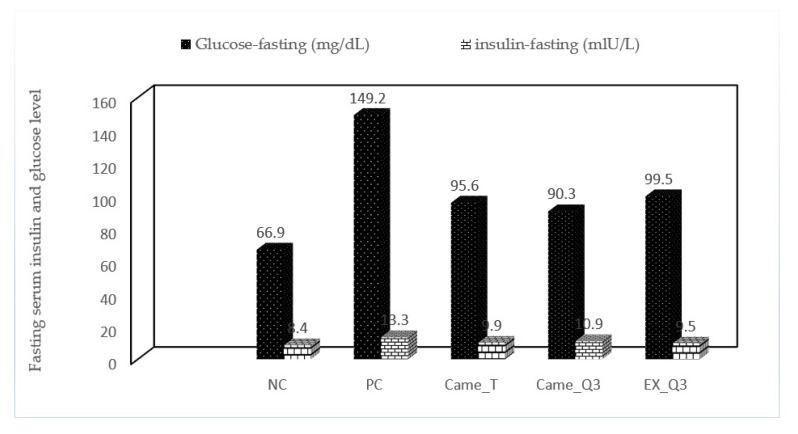
Effect of fermented camel milk fortified with red quinoa seed powder on fasting serum insulin and glucose levels in hypercholesterolemic rats.

**Figure 3 molecules-27-07695-f003:**
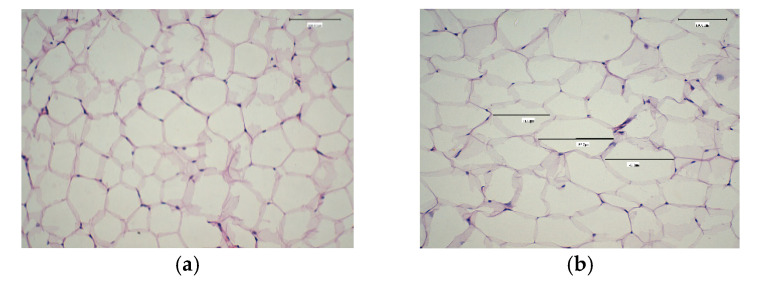
Effect of fermented camel milk either alone or fortified with 3% of quinoa flour on adipocyte cells. Adipocytes in paraffin sections: (scale bar, 50 μm, magnification, 100×). (**a**) Negative control group. (**b**) Positive control group. (**c**) Rats treated with fermented camel milk. (**d**) Rats treated with fermented camel milk fortified with 3% quinoa flour.

**Figure 4 molecules-27-07695-f004:**
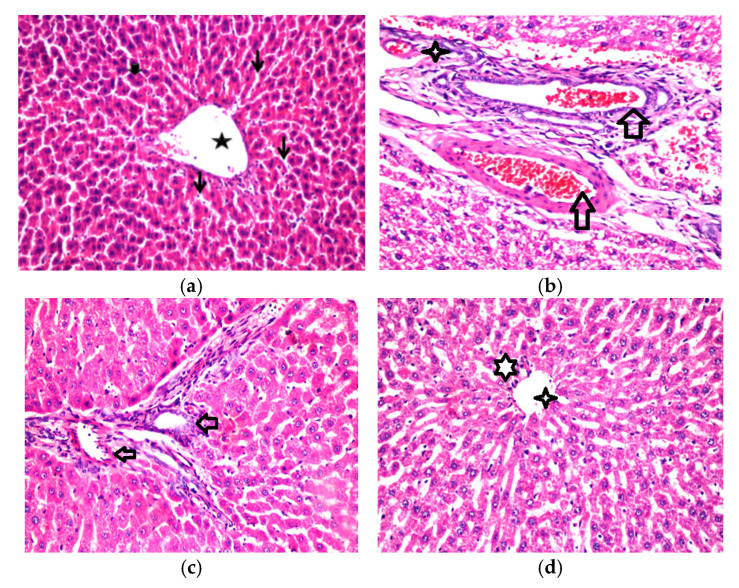
Effect of fermented camel milk fortified with red quinoa seed flour on hepatic tissue in in hypercholesterolemic rats. (**a**) Negative control group (Representative photomicrograph of rat liver showing normal histo-morphological structures including central vein (star), hepatic cords (thick arrow) and sinusoids (thin arrows). H&E stain magnification ×100). (**b**) Positive control group (Representative photomicrograph of rat liver showing mild congested blood vessels and sinusoids (star), mild Kupffer cells hyperplasia (small arrow) besides mild lymphocytic infiltrations (arrows). H and E stain magnification ×200). (**c**) Rats treated with fermented camel milk (Representative photo-micrograph of rat liver showing slight portal fibrosis among proliferated and new formed bile ductless beside peri-artery inflammatory cells aggregations mainly lymphocytes (arrow). H and E stain magnification ×200). (**d**) Rats treated with fermented camel milk fortified with 3% of quinoa flour (Representative photomicrograph of rat liver showing nearly normal hepatic lobules with mild congested central vein (star). H and E stain magnification ×100).

**Figure 5 molecules-27-07695-f005:**
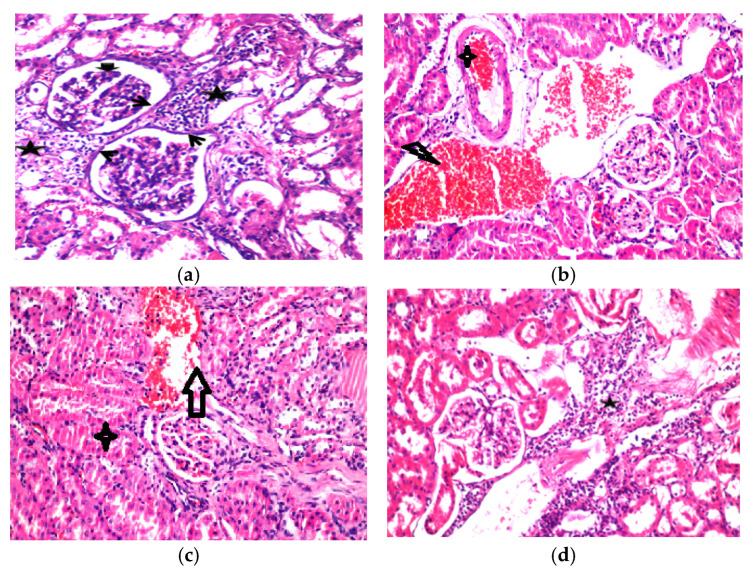
Effect of fermented camel milk fortified with red quinoa seed powder on the histological examination of the kidneys in hypercholesterolemic rats. (**a**) Negative control group (Representative photomicrograph of rat kidney showing normal histological architectures including glomeruli (stars) and tubules (arrows). H and E stain magnification ×100). (**b**) Positive control group (Representative photomicrograph of rat kidney showing renal cortex with areas of large eosinophilic casts (stars) beside degenerated tubules (small arrows) and partial shrunken glomeruli (arrow) with widening of bowman’s spaces. H and E stain magnification ×100). (**c**) Rats treated with fermented camel milk (Representative photomicrograph of rat kidney showing apparently normal glomeruli and tubules except mild congested large interstitial renal blood vessels (star) beside degenerated proximal tubules (arrow). H and E stain magnification ×200). (**d**) Rats treated with fermented camel milk fortified with 3% quinoa flour (Representative photomicrograph of rat kidney showing marked interstitial lymphocytic infiltrations (star) besides nearly normal glomeruli. H and E stain magnification ×200).

**Figure 6 molecules-27-07695-f006:**
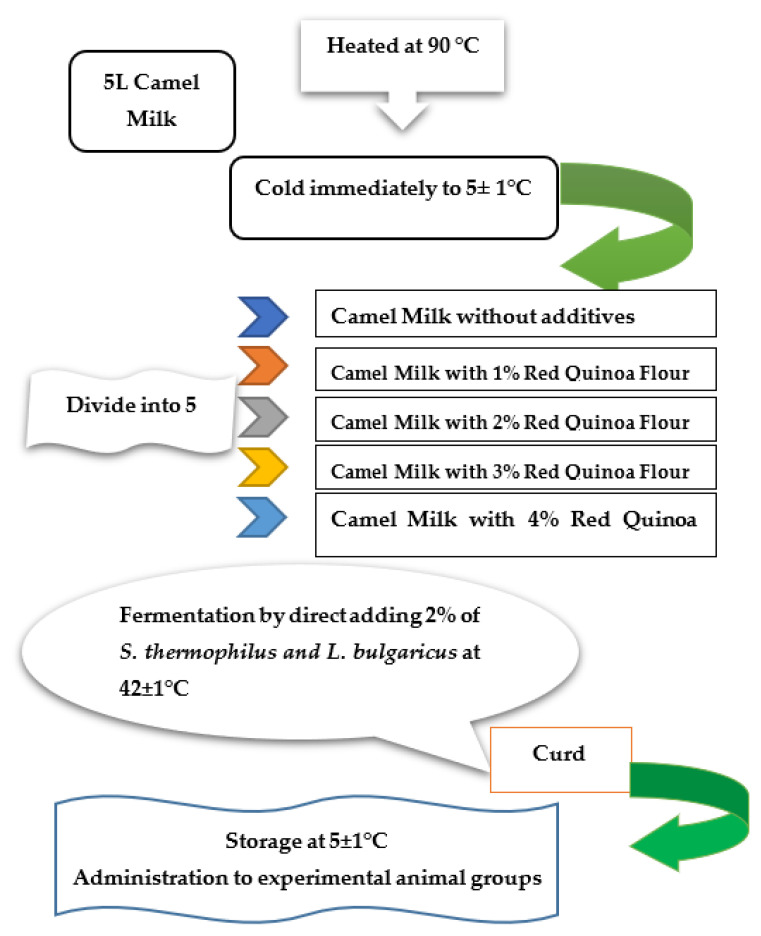
Basic process for manufacturing fermented camel fortified with red quinoa four.

**Table 1 molecules-27-07695-t001:** (**a**). Chemical composition and energy value of red quinoa seed. (**b**). Content of chosen mineral in red quinoa seed. (**c**). Yield extract, polyphenol compounds and total antioxidant capacity in red quinoa seed.

(**a**)	
**Chemical Composition**	**Chemical Composition (g/100g) on Dry Weight**
Protein (%)	15.0 ± 0.24
Fiber (%)	8.8 ± 0.05
Ash (%)	3.2 ± 0.2
Carbohydrates (%)	57.58 ± 0.21
Fat (%)	5.2 ± 0.14
Moisture (10)	10.22 ± 0.21
Energy (kcal/100g)	352.12 ± 0.7
(**b**)	
**Chemical Composition**	**Eliminates Instead of Mineral (mg/100 g)**
Calcium	148.7 ± 1.2
Phosphorous	383.9 ± 2.3
Magnesium	246.9 ± 2.2
Sodium	12.2 ± 0.2
Potassium	926.7 ± 4
Iron	13.2 ± 0.3
Manganese	10.1 ± 0.24
Copper	5.1 ± 0.2
Zinc	4.4 ± 0.16
(**c**)	
**Phytochemical Properties**	
Yield g/100 g DW Vitamin C (mg/100 g DW)	16.86 ± 1.5 16.4 ± 0.24
Phenolic compounds (mg GAE/100 g DW)	488.32 ± 2.91
Total Flavonoid (mg CE/100 g DW)	367.11 ± 4.22
DPPH* Inhibition (%)	47.0 ± 2.4

Data are presented as the mean ± SD (*n* = 3/group). GAE: gallic acid equivalents; CE: catechin equivalents; DW: dry weight. The mean and standard deviation (*n* = 3) are reported.

**Table 2 molecules-27-07695-t002:** (**a**). Physicochemical of different fermented milks fortified with red quinoa seeds. (**b**). Sensory evaluation of different fermented milks fortified with red quinoa seeds.

(**a**)					
**Attribute**	**Fermented Milk Types**				
	**Cam_T**	**Cam_Q_1_**	**Cam_Q_2_**	**Cam_Q_3_**	**Cam_Q_4_**
Total Solids (%)	14.60 ^e^ ± 0.55	15.30 ^d^ ± 0.48	16.14 ^c^ ± 0.68	17.02 ^b^ ± 0.88	17.98 ^a^ ± 0.78
Protein (%)	5.02 ^c^ ± 0.16	5.14 ^bc^ ± 0.20	5.26 ^b^ ± 0. 24	5.36 ^ab^ ± 0. 22	5.44 ^a^ ± 0.24
Fat (%)	4.04 ^a^ ± 0.12	4.18 ^a^ ± 0.16	4.24 ^a^ ± 0.14	4.30 ^a^ ± 0.16	4.38 ^a^ ± 0.18
Ash (%)	0.96 ^a^ ± 0.09	1.00 ^a^ ± 0.07	1.05 ^a^ ± 0.10	1.09 ^a^ ± 0.08	1.14 ^a^ ± 0.05
Fiber (%)	0.00	0.12 ^d^ ± 0.02	0.23 ^c^ ± 0.03	0.34 ^b^ ± 0.02	0.46 ^a^ ± 0.03
Acidity (%)	0.78 ^c^ ± 0.04	0.82 ^b^ ± 0.03	0.88 ^ab^ ± 0.06	0.90 ^a^ ± 0.02	0.92 ^a^ ± 0.02
pH	4.80 ^a^ ± 0.08	4.76 ^b^ ± 0.06	4.70 ^c^ ± 0.09	4.66 ^d^ ± 0.06	4.62 ^e^ ±0.05
(**b**)					
**Sensory Properties**	**Fermented Milk Types**				
	**Cam_T**	**Cam_Q_1_**	**Cam_Q_2_**	**Cam_Q_3_**	**Cam_Q_4_**
Flavor (30)	21.18 ^e^ ± 1.2	25.30 ^c^ ± 1.4	26.60 ^b^ ± 1.3	27.10 ^a^ ± 1.1	24.60 ^d^ ± 1.3
Color (10)	30.20 ^e^ ± 4.5	32.24 ^d^ ± 3.7	34.2 ^b^ ± 4.6	35.40 ^a^ ± 4.92	36.20 ^c^ ± 3.8
Body and Texture (40)	8.2 ^d^ ± 0.3	8.6 ^a^ ± 0.4	8.4 ^b^ ± 0.4	8.3 ^c^ ± 0.6	7.9 ^e^ ± 0.5
Acidity (10)	8.82 ^c^ ± 0.2	8.93 ^bc^ ± 0.2	9.02 ^b^ ± 0.2	9.17 ^a^ ± 0.1	8.58 ^d^ ± 0.2
Overall acceptability (10)	9.01 ^bc^ ± 0.1	9.11 ^b^ ± 0.1	9.14 ^ab^ ± 0.2	9.25 ^a^ ± 0.1	8.85 ^c^ ± 0.2
Total (100)	77.41 ^d^ ± 3.12	84.18 ^c^ ± 4.22	87.36 ^b^ ± 3.70	89.22 ^a^ ± 3.82	86.13 ^ab^ ± 3.70

Data are presented as the mean ± SD (*n* = 3/group). Different superscript letters (a to e) within the same raw showed significant differences among the groups (*p* ≤ 0.05). Fermented camel milk as a control (Cam_T), Fermented camel milk fortified with 1% red quinoa seed flour (Cam_Q_1_), Fermented camel milk fortified with 2% red quinoa seed powder (Cam_Q_2_), Fermented camel milk fortified with 3% red quinoa seed flour (Cam_Q_3_) and Fermented camel milk fortified with 4% red quinoa seed flour (Cam_Q_4_).

**Table 3 molecules-27-07695-t003:** Effect of fermented camel milk fortified with red quinoa seed flour on the lipid profile and atherogenic index in hypercholesterolemic rats.

Groups	TG mg/dL	TC mg/dL	HDL mg/dL	LDL mg/dL	VLDL mg/dL
NC	65.9 ^d^ ± 1.1	64.9 ^d^ ± 2.7	37.2 ^a^ ± 0.9	14.5 ^e^ ± 1.3	13.2 ^c^ ± 0.1
PC	146.3 ^a^ ± 5.7	152.4 ^a^ ± 6.2	24.5 ^d^ ± 0.8	98.5 ^a^ ± 5.8	29.3 ^a^ ± 1.1
Cam_T	97.7 ^b^ ± 2.3	105.1 ^b^ ± 5.6	32.6 ^b^ ± 0.8	53.0 ^b^ ± 3.4	19.5 ^b^ ± 0.6
Cam_Q_3_	93.6 ^c^ ± 1.9	96.1 ^c^ ± 4.6	36.2 ^ab^ ± 0.9	41.2 ^d^ ± 2.7	18.7 ^bc^ ± 0.5
EX_Q_3_	96.4 ^b^ ± 2.1	99.3 ^bc^ ± 5.1	30.5 ^cd^ ± 0.8	49.3 ^c^ ± 2.9	19.3 ^b^ ± 0.6

Data are presented as the mean ± SD (*n* = 3/group). Different superscript letters (a to e) within the same raw showed significant differences among the groups (*p* ≤ 0.05). (NC) Nontreated nonhypercholesterolemic rats (negative control), (PC) hypercholesterolemic rats (positive control), (Cam_T) hypercholesterolemic rats treated with fermented camel milk, and (Cam_Q_3_) hypercholesterolemic rats treated with fermented camel milk fortified with 3% red quinoa seed flour and (EX_Q_3_) aqueous extract of red quinoa seeds flour.

**Table 4 molecules-27-07695-t004:** Effect of different fermented milks on albumin, protein, serum enzyme activities of liver, and kidney markers in hypercholesterolemic rats.

Groups	ALP (U/L)	ALT (U/L)	AST (U/L)	Total Protein (g/dL)	Albumin (g/dL)	Urea (mg/dL)	Creatinine (mg/dL)
NC	105.2 ^b^ ± 7.1	25.2 ^d^ ± 1.74	52.3 ^d^± 5.4	7.3 ^b^ ± 0.1	3.85 ^a^ ± 0.04	17.54 ^d^ ± 0.1	0.89 ^e^ ± 0.3
PC	180.4 ^a^ ± 9.4	58.8 ^a^ ± 3.72	115.2 ^a^ ± 6.9	5.5 ^c^ ± 0.11	2.85 ^c^ ± 0.01	40.89 ^a^ ± 0.3	2.4 ^a^ ± 0.12
Cam_T	141.5 ^c^ ± 6.8	36.4 ^c^ ± 1.80	79.8 ^b^ ± 5.12	7.3 ^bc^ ± 0.12	3.81 ^ab^ ± 0.02	22.1 ^b^ ± 0.13	1.96 ^b^ ± 0.5
Cam_Q_3_	128.6 ^d^ ± 5.7	33.8 ^c^ ± 3.18	73.2 ^c^ ± 2.7	7.6 ^b^ ± 0.09	4.15 ^a^ ± 0.03	19.75 ^c^ ± 0.9	1. 5 ^d^ ± 0.11
EX_Q_3_	135.6 ^cd^ ± 5.7	41.7 ^b^ ± 1.2	84.3 ^b^ ± 5.2	7.9 ^a^ ± 0.08	3.70 ^b^ ± 0.04	23.1 ^b^ ± 0.19	1.75 ^c^ ± 0.1

Data are presented as the mean ± SD (*n* = 3/group). Different superscript letters (a–d) within the same raw showed significant differences among the groups (*p* ≤ 0.05). (NC) Nontreated nonhypercholesterolemic rats (negative control), (PC) hypercholesterolemic rats (positive control), (Cam_T) hypercholesterolemic rats treated with fermented camel milk, (Cam_Q3) hypercholesterolemic rats treated with fermented camel milk fortified with 3% red quinoa seed flour and (EX_Q3) aqueous extract of red quinoa seeds.

**Table 5 molecules-27-07695-t005:** Effect of fermented camel milk fortified with red quinoa seed flour on antioxidant markers in hypercholesterolemic rats.

Groups	MDA (μmol/L)	TAC (μmol/L)	SOD (U/g Hb)	GSH-px (U/g Hb)
NC	1.27 ^d^ ± 0.05	780 ^a^ ± 14	5.34 ^a^ ± 0.85	22.14 ^a^ ± 0.88
PC	2.13 ^a^ ± 0.07	550 ^e^ ± 21	3.28 ^d^ ± 0.40	15.12 ^d^ ± 0.76
Cam_T	1.56 ^b^ ± 0.03	655 ^d^ ± 18	4.08 ^c^ ± 0.55	17.28 ^c^ ± 0.85
Cam_Q_3_	1.32 ^c^ ± 0.02	775 ^b^ ± 16	4.60 ^b^ ± 0.68	18.22 ^b^ ± 0.80
EX_Q_3_	1.30 ^cd^ ± 0.01	735 ^c^ ± 15	3.98 ^c^ ± 0.55	16.82 ^c^ ± 0.85

Data are presented as the mean ± SD (*n* = 3/group). Different superscript letters (a–e) within the same raw showed significant differences among the groups (*p* ≤ 0.05). (NC) nontreated nonhypercholesterolemic rats (negative control), (PC) hypercholesterolemic rats (positive control), (Cam_T) hypercholesterolemic rats treated with fermented camel milk, (Cam_Q3) hypercholesterolemic rats treated with fermented camel milk fortified with 3% red quinoa seed flour and (EX_Q3) aqueous extract of red quinoa seeds.

**Table 6 molecules-27-07695-t006:** Effect of fermented camel milk fortified with red quinoa seed flour on liver weight, adipose tissue, and fat cell volume in hypercholesterolemic rats.

Groups	Liver (g)	White Adipose Tissue (g)	Adipocyte Size (μm^2^ × 10^3^)
NC	8.59 ^b^ ± 0.2	1.66 ^d^ ± 0.2	3.1 ^d^ ± 0.4
PC	9.99 ^a^ ± 0.3	5.45 ^a^ ± 0.6	10.96 ^a^ ± 0.7
Cam_T	7.9 ^cd^ ± 0.8	2.72 ^b^ ± 0.5	6.3 ^b^ ± 0.4
Cam_Q_3_	8.1 ^c^ ± 0.3	2.3 ^c^ ± 0.4	4.55 ^c^ ± 0.2
EX_Q_3_	7.5 ^d^ ± 0.8	2.62 ^b^ ± 0.5	5.9 ^bc^ ± 0.4

Data are presented as the mean ± SD (*n* = 3/group). Different superscript letters (a to d) within the same raw showed significant differences among the groups (*p* ≤ 0.05). (NC) nontreated nonhypercholesterolemic rats (negative control), (PC) hypercholesterolemic rats (positive control), (Cam_T) hypercholesterolemic rats treated with fermented camel milk, (Cam_Q3) hypercholesterolemic rats treated with fermented camel milk fortified with 3% red quinoa seed flour and (EX_Q3) aqueous extract of red quinoa seeds.

## Data Availability

The authors declare the availability of data and material; they also declare the data is transparent for this manuscript.

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
