# Peer review of "Nutritional and Therapeutic Properties of Fermented Camel Milk Fortified with Red *Chenopodium quinoa* Flour on Hypercholesterolemia Rats"

_molecules, 2022, doi:10.3390/molecules27227695_

Round 1
Reviewer 1 Report
why choose to use fermented camel milk over fresh camel milk?
there are some jumps in the introduction between camel milk and red Chenopodium quinoa seeds and Diseases not Consistent the Writing should be improved
the chemical component of camel milk fresh and fermented should be demonstrated
the effect of adding quina flour to the milk on the chemical composition should be conducted
Author Response
Dear Sir,
Thank you for your useful comments and suggestions on the language and structure of our manuscript. We have modified the manuscript accordingly, and detailed corrections are listed below point by point.
|
First Reviewer Comments |
||
|
Comments |
Responses |
Page/Line |
|
Why choose to use fermented camel milk over fresh camel milk?
|
Because fermented milk contains all the biological compounds of fresh camel milk, including antioxidants, healthy fats, like fatty acids, linoleic acid, and unsaturated fatty acids, which may support brain and heart health, as well as more vitamin C and vitamin B, calcium, iron, and potassium. Additionally, it contains peptides that are produced during fermentation and support a higher level of the body's immunity, making it the ideal choice for people who have from lactose intolerance or milk allergy. |
-- |
|
There are some jumps in the introduction between camel milk and red Chenopodium quinoa seeds and Diseases are not Consistent the Writing should be improved
|
Done |
Page 2 Lines 6-37 |
|
The chemical composition of camel milk fresh and fermented should be demonstrated
|
Done
|
Page 4 Lines 6-16 |
|
The effect of adding quinoa flour to the milk on the chemical composition should be conducted.
|
The impact of adding quinoa flour to camel milk was investigated since it raised the percentage of lactose, which sped up the fermentation process in comparison to camel milk without adding it. The milk's content also increased in terms of dietary fiber and protein (results not shown). |
-- |

Reviewer 2 Report
Abstract:
1- It is in need to redesign with logic flow of steps and providing the meaning of abbreviated terms.
2- Revise the word “fortifying” line 3
Key words are very broad and need some specific words related to results
Introduction:
1- revise the word “Arab Gulf” line 2
2- the sentence “Beneficial uses of camel milk in the metabolic syndrome (which is the combination of abdominal obesity, dyslipoproteinemia (de-creased HDL and increased VLDL), and hypertension with peripheral insulin resistance as the causal defect) have frequently been postulated [1].” Is very lengthy and need to be rephrased.
3- The meaning of any abbreviations must be given at the first appearance in the manuscript. For example “, LDL, LAB” even if it is very common term.
4- the authors specified the aim of work at the end of introduction “The purpose of this study is to determine the hepatoprotective effects of fermented camel milk supplemented with red quinoa flour on lipid profile, biochemical markers, and histopathological characteris-tics in hypercholesterolemia rats.”
But the work included many other parametrs including Nutritional properties of fermented camel milk and effect on kidney….etc. it is better to mention those goals
Results:
For the section 2.1
1-line 4 revise the writing of “previous studies by [21,22]” to be expressed as author name et al., [21,22].
2- It is recommended to divide table 1 into separate tables each table to discuss various properties because each property should be expressed by different way “like %, amount in mg/ 100 g DW) and it is incorrect to involve all in one table. And give the meaning of abbreviations after the table “the legends”
For the section 2.2
1- The authors selected the (3% v/w) as an ideal ratio to be used and give it the abbreviation 3% of quinoa flour (Cam_Q3), and then mentioned the abbreviation of other concentrations in table 2 only. So, recommended to mention the abbreviation of other concentrations “1, 2, and 4 %" also in the discussion. Despite the values of Physicochemical and Sensory evaluation all concentrations are very close they selected only 3% , WHY?
2- Table 2 needs to be redesigned.
3- What is the meaning of TS in table 2?
4- The authors used in legends for Cam_Q1, Q3 and Q4 seed flour while for Q2 used seed powder could they explain the difference?
For the section 2.3
1- (EX_Q3) aqueous solution of red quinoa seeds flour is mentioned but it is not described in methods and why the authors used it. Please explain and mention.
2- correct the number of figure it is figure 1 not 2.
For the section 2.5
1- Provide the full meaning of all abbreviations “TC, TG, LDL and VLDL,………and the other remaining” at the first appearance.
2- The units in table 3 “mg/dl” make it to be “mg/dL “for the uniformity of manuscript.
For the section 2.6
1- Provide the full meaning of all abbreviations “AST, ALP, ALT,…etc ” at the first appearance.
2- revise the units of measurements of various parameters for example” ALT is in IU/L, ALP in U/L and AST in IU/L”
3-“mg/dl” make it to be “mg/dL “for the uniformity of manuscript
For the section 2.7
1- Provide the full meaning of all abbreviations
2- Rearrange the photos and its corresponding titles because it is confusing for the readers.
For the section 2.8 Histological changes in experimental rats fed on fermented camel milk fortified with red quinoa.
This section is in need for redesign of numbering the figures and separate it into different organs” each organ to be in a separate figure” very confusing and unusual way of expressing the data.
3. Material and Methods
- The methods of incorporation of Quinoa during the preparation of yogurt is not mentioned or it is missed please, revise.
- Section 3.7 is in italic form please check.
- Figure of Basic process for manufacturing fermented camel fortified with red quinoa four must be provided in better form and resolution and give it the correct number according to the flow of numbers of figures “
Author Response
Dear Sir,
Thank you for your useful comments and suggestions on the language and structure of our manuscript. We have modified the manuscript accordingly, and detailed corrections are listed below point by point. We have revised the whole manuscript carefully and tried to avoid any grammar or syntax errors. We believe that the language is now acceptable for the review process. We standardized the names and abbreviations of the microorganisms in all parts of the manuscript and wrote them in italics. We have re-designed and improved the quality and accuracy of the figures to suit the journal's format and to become more homogeneous. All figures and tables are provided and cited in sequence in the main text. We have checked all the references and formatted them strictly according to the Guide for Authors.
|
Second Reviewer Comments |
||
|
Comments |
Responses |
Page/Line |
|
Abstract: |
||
|
1-It is in needs to redesign with logic flow of steps and providing the meaning of abbreviated terms.
|
Done |
Page 1
|
|
2- Revise the word “fortifying” in line 3
|
Done |
Page 1
|
|
Keywords are very broad and need some specific words related to the results |
Done |
Page 1
|
|
Introduction: |
||
|
1- revise the word “Arab Gulf” in line 2 |
Done |
Page 1 |
|
2- the sentence “Beneficial uses of camel milk in the metabolic syndrome (which is the combination of abdominal obesity, dyslipoproteinemia (de-creased HDL and increased VLDL), and hypertension with peripheral insulin resistance as the causal defect) have frequently been postulated [1].” Is very lengthy and needs to be rephrased. |
Done |
Page 1 |
|
3-The meaning of any abbreviations must be given at the first appearance in the manuscript. For example “, LDL, LAB” even if it is very common term. |
Done |
Page 2 |
|
4-the authors specified the aim of work at the end of introduction “The purpose of this study is to determine the hepatoprotective effects of fermented camel milk supplemented with red quinoa flour on lipid profile, biochemical markers, and histopathological characteristics in hypercholesterolemia rats.” But the work included many other parameters including Nutritional properties of fermented camel milk and effect on kidney….etc. it is better to mention those goals |
Done |
Page 2 |
|
Material and Methods |
||
|
The methods of incorporation of Quinoa during the preparation of yogurt is not mentioned or it is missed please, revise. |
Done |
Page 14 |
|
Figure 1. of the Basic process for manufacturing fermented camel fortified with red quinoa four must be provided in better form and resolution and give it the correct number according to the flow of numbers of figures “. |
Done |
Page 16 |
|
Section 3.7 is in the italic form please check. |
Done |
Page 14 |
|
Results |
||
|
For the section 2.1 1-line 4 revise the writing of “previous studies by [21,22]” to be expressed as author name et al., [21,22]. |
Done |
|
|
2- It is recommended to divide table 1 into separate tables each table to discuss various properties because each property should be expressed by different way “like %, amount in mg/ 100 g DW) and it is incorrect to involve all in one table. And give the meaning of abbreviations after the table “the legends” |
Done |
|
|
For the section 2.2 1-The authors selected the (3% v/w) as an ideal ratio to be used and give it the abbreviation 3% of quinoa flour (Cam_Q3), and then mentioned the abbreviation of other concentrations in table 2 only. So, recommended mentioning the abbreviation of other concentrations “1, 2, and 4 %" also in the discussion. Despite the values of Physicochemical and Sensory evaluation, all concentrations are very close they selected only 3%, WHY?
|
The samples manufactured from fermented camel milk fortified with 3% red quinoa seed flour recorded the best flavour, colour, taste, and general acceptance, and the fermented product had high levels of antioxidants (phenolic compounds, flavonoids, and vitamin C) compared to the mixing ratios of 1 and 2%. As for fermented milk fortified with 4% of red quinoa, which received the worst sensory rating compared to other types, despite containing higher levels of antioxidants, and therefore we may not be able to produce it on a commercial scale. |
|
|
2- Table 2 needs to be redesigned.
|
Done |
|
|
3- What is the meaning of TS in table 2? |
It mean total solids |
|
|
4- The authors used in legends for Cam_Q1, Q3, and Q4 seed flour while for Q2 used seed powder could they explain the difference? |
Done |
|
|
For the section 2.3 1-(EX_Q3) aqueous solution of red quinoa seeds flour is mentioned but it is not described in the methods and why the authors used it. Please explain and mention. |
Done |
Page 14 |
|
2- correct the number of figure it is figure 1 not 2. |
Done |
|
|
For the section 2.5 1- Provide the full meaning of all abbreviations “TC, TG, LDL and VLDL,………and the other remaining” at the first appearance |
Done |
|
|
2-The units in table 3 “mg/dl” make it to be “mg/dL “for the uniformity of the manuscript. |
Done |
|
|
For the section 2.6 1-Provide the full meaning of all abbreviations “AST, ALP, ALT,…etc ” at the first appearance. |
Done |
Page 8-9 |
|
2-revise the units of measurements of various parameters for example” ALT is in IU/L, ALP in U/L and AST in IU/L” |
Done |
Page 9 |
|
3-“mg/dl” make it to be “mg/dL “for the uniformity of manuscript |
Done |
|
|
For section 2.7 1-Provide the full meaning of all abbreviations |
Done |
|
|
2- Rearrange the photos and its corresponding titles because it is confusing for the readers. |
Done |
Page 9-13 |
|
For section 2.8 Histological changes in experimental rats fed on fermented camel milk fortified with red quinoa. This section is in need of redesign of numbering the figures and separating them into different organs” each organ to be in a separate figure” very confusing and unusual way of expressing the data. |
Done |
Page 12-13 |
Reviewer 3 Report
The authors evaluated the Nutritional and therapeutic properties of fermented camel milk fortified with red Chenopodium quinoa flour on hypercholesterolemia rats.
The paper is well written and well organized.
Some minor remarks are following
2.2. Gross chemical composition and Sensory....Not capital the S
Table 2. Physicochemical and Sensory evaluation of different fermented milks...The same
Chenopodiaceae family...Chenopodiaceae not in italics. Please check in the whole text.
3.7. Chemical analysis... The first paragraph not in italics
Figure 1...What are all those black areas?
Please write all references in a uniform manner and according to the guidelines of the journal.
Author Response
Dear Sir,
Thank you for your useful comments and suggestions on the language and structure of our manuscript. We have modified the manuscript accordingly, and detailed corrections are listed below point by point. We have re-designed and improved the quality and accuracy of the figures to suit the journal's format and to become more homogeneous. All figures and tables are provided and cited in sequence in the main text. We have checked all the references and formatted them strictly according to the Guide for Authors.
|
Third Reviewer Comments |
||
|
Comments |
Responses |
Line |
|
2.2. Gross chemical composition and Sensory....Not capital the S |
Done |
|
|
Table 2. Physicochemical and Sensory evaluation of different fermented milks...The same |
Done |
|
|
Chenopodiaceae family...Chenopodiaceae not in italics. Please check in the whole text. |
Done |
|
|
3.7. Chemical analysis... The first paragraph not in italics |
Done |
|
|
Figure 1...What are all those black areas? |
Done |
|

Round 2
Reviewer 2 Report
The authors improved the quality of manuscript. All the best.
Author Response
Dear Sir,
Thank you for your useful comments on the English language and style. We have modified the manuscript accordingly and carefully corrected the language and style to avoid any grammar, spelling, or syntax errors.
We believe that the language is now acceptable for the publisher process.
Sincerely,
Dr. Khaled M. ElZahar
